# Newborn Screening for CF across the Globe—*Where Is It Worthwhile*?

**DOI:** 10.3390/ijns6010018

**Published:** 2020-03-04

**Authors:** Virginie Scotet, Hector Gutierrez, Philip M. Farrell

**Affiliations:** 1Inserm, University of Brest, EFS, UMR 1078, GGB, F-29200 Brest, France; 2Department of Pediatrics, University of Alabama at Birmingham, Birmingham, AL 35233, USA; hgutierrez@peds.uab.edu; 3Departments of Pediatrics and Population Health Sciences, University of Wisconsin School of Medicine and Public Health, Madison, WI 53705, USA; pmfarrell@wisc.edu

**Keywords:** cystic fibrosis, newborn screening, incidence, malnutrition, cost, health policy

## Abstract

Newborn screening (NBS) for cystic fibrosis (CF) has been performed in many countries for as long as four decades and has transformed the routine method for diagnosing this genetic disease and improved the quality and quantity of life for people with this potentially fatal disorder. Each region has typically undertaken CF NBS after analysis of the advantages, costs, and challenges, particularly regarding the relationship of benefits to risks. The very fact that all regions that began screening for CF have continued their programs implies that public health and clinical leaders consider early diagnosis through screening to be *worthwhile.* Currently, many regions where CF NBS has not yet been introduced are considering options and in some situations negotiating with healthcare authorities as policy and economic factors are being debated. To consider the assigned question (*where is it worthwhile*?), we have completed a worldwide analysis of data and factors that should be considered when CF NBS is being contemplated. This article describes the lessons learned from the journey toward universal screening wherever CF is prevalent and an analytical framework for application in those undecided regions. In fact, the lessons learned provide insights about what is necessary to make CF NBS *worthwhile*.

## 1. Introduction

To appreciate what makes cystic fibrosis (CF) newborn screening (NBS) *worthwhile*, if not essential, it is helpful to review briefly certain historical aspects and thereby supplement the overall history described herein by Travert [1]. In particular, the perspective that follows focuses on the lessons learned about what is needed to ensure that early diagnosis through screening is indeed *worthwhile* for individuals and targeted populations. According to the Cambridge English Dictionary, *worthwhile* means “useful, important, or good enough to be a suitable reward for the money or time spent.” Currently, the majority of countries in Europe and those elsewhere populated by inhabitants with European ancestry are screening newborns for CF, as shown in Figure 1. Each of these regions faced and overcame many challenges such as those listed in Table 1. Often, the combination of laboratory difficulties and complicated but necessarily efficient follow-up systems proved daunting. It may be assumed that all these regions consider CF NBS *worthwhile.* Of course, CF NBS will not prove *worthwhile* unless sustained financial support can be anticipated and all of the essential elements shown in Figure 2 are well organized and maintained. Experience has shown that the NBS system of early diagnosis and treatment requires that every step in the process be performed with assured high quality. 

## 2. Requirements That Must Be Met for CF NBS to Be *Worthwhile*

### 2.1. Feasibility of Screening Newborns for CF

The requirements that must be addressed to implement and maintain a successful NBS program for CF are listed in Table 1. Although attempts to achieve early diagnosis of CF through meconium tests were organized during the 1970s [2], NBS first became feasible on a population scale in 1979 when dried blood spots were analyzed for immunoreactive trypsinogen (IRT) in New Zealand by Crossley et al. [3]. Through retrospective assessment, they found that high IRT levels revealed a significant risk for CF. The utility and convenience of dried blood spots in NBS had, of course, been obvious since their application in 1963 to phenylketonuria [4], and many public health laboratories worldwide were already screening for hereditary metabolic disorders and congenital endocrinopathies. Thus, the first requirement for the *worthwhile* implementation of CF NBS is to have a system in place and functioning well for the universal collection of dried blood spot specimens and their analysis in a central laboratory with quality assurance mechanisms in place. The seminal research in New Zealand was only possible because the NBS laboratory there collaborated closely with the University of Auckland Paediatrics Department across the street. The lesson learned there has been demonstrated repeatedly, namely that to be *worthwhile*, CF NBS must be a collaborative effort with a dedicated team that addresses every component of the sequence shown in Figure 2. For those few engaged in CF NBS using meconium tests in the 1970s, the report of Crossley et al. [3] had an immediate, profound influence, stimulating research around the world. On the other hand, the availability of a screening test alone is insufficient to justify its implementation as Wilson and Jungner [5] emphasized five decades ago. 

### 2.2. The Need for an Excellent Screening Test: Limitations of IRT/IRT

The breakthrough discovery in New Zealand was followed by important studies in New South Wales in Australia [6], Colorado [7], and France [8]. Leaders in each of these regions recognized the potential benefits of early diagnosis ranging from epidemiologic and clinical research opportunities to care enhancement and improvement in the organization of healthcare delivery. Much skepticism remained, however, because of concerns about the IRT test per se, whether or not significant clinical benefits actually occurred, and how much adverse impact was being imposed on parents of screened neonates, i.e., the degree of psychosocial harm. In retrospect, the major concern that limited CF NBS acceptance, and thus a third lesson learned, concerns the IRT/IRT screening strategy—a method with relatively low sensitivity that requires a second, confirming blood specimen at approximately two weeks of age. Consequently, this first phase of experience with dried blood spot screening led to a realization that more research was needed on all aspects of CF NBS and the IRT method of screening needed to be improved. In fact, a decade after the report from New Zealand there was worldwide debate among health policy decision-makers whether or not CF NBS was *worthwhile* and even doubt among organizations like the U.S. Cystic Fibrosis Foundation—expressed emphatically when it sponsored a negative but influential commentary [9]. Consequently, CF NBS implementation was slow in North America and Europe, and one country (France) even discontinued their national IRT-based program. 

### 2.3. The Value of the IRT/DNA Screening Test When CFTR Mutations Are Known

The view that IRT/IRT was not sufficiently sensitive with practical cutoff values coupled to the discovery of the *CFTR* gene in 1989 [10] and its principal disease-causing variant, p.Phe508del (F508del), led almost immediately to a search for a better screening algorithm in some regions. Others, however, continued IRT/IRT and either tolerated, or did not recognize, its relatively low sensitivity of 75–80% [11]. In retrospect, the discovery that about 90% of Europeans and Europe-derived CF populations have at least one p.Phe508del variant greatly facilitated the development of the first DNA-based NBS test, the IRT/DNA(p.Phe508del) method [12]. Soon thereafter, the DNA tier was expanded to a *CFTR* multimutation panel and a sensitivity of >95% was achieved routinely [13]. In addition, the quality of screening improved significantly by allowing test completion on the initial dried blood spot specimen, thus improving timeliness, and by providing valuable information on *CFTR* mutations. It was quickly learned with IRT/DNA(*CFTR*) that the vast majority of CF cases can be presumptively (genetically) diagnosed within a week of birth from the initial blood specimen and valuable genetic data obtained to predict pancreatic functional status. 

The lesson learned from these experiences is clear: although initiating CF NBS with the IRT biomarker alone is much better than no screening for CF, regions should plan from the outset on improving their laboratory methods, ideally with a DNA-based second-tier method as *CFTR* population data emerge and enable transformation to a better, DNA-based algorithm that can make CF NBS more *worthwhile.* Another alternative is to use pancreatitis-associated protein (PAP) as an adjunct but a variety of issues limit its effectiveness [14]. The motivation for including PAP as a secondary biomarker in the screening strategy was to limit the incidental findings inherent to the use of DNA analysis such as the detection of carriers and the recognition of equivocal clinical phenotypes [14]. With any algorithm, tracking and evaluating data annually is an essential component of monitoring screening outcome measures such as sensitivity, specificity, positive predictive value, age of diagnosis to ensure that timeliness is achieved and disease incidence. 

### 2.4. The Challenge of Evaluating and Achieving Benefits That Outweigh Risks 

No NBS test has been subjected to more skepticism or scrutiny than CF screening even when the unique value of the IRT/DNA screening became evident. The rationale illustrated in Figure 3 has been considered so intuitive for other genetic conditions that implementation with little or no clinical evidence is typical for most screening tests and cost-effectiveness is generally not assessed. In retrospect, the debates in the CF community that raged over whether or not actual benefits occur and, if so, the benefit: risk relationship seems surprising when the potentially fatal salt loss in sweat and protein-energy malnutrition are well known to have plagued children with CF for decades [15]. Thus, the proof was demanded for the efficacy of CF NBS. However, the very short pre-symptomatic phase of CF, illustrated in Figure 3, was not appreciated until studies of infants diagnosed through NBS were published and revealed that malnutrition may occur within days and lung disease within weeks [15]. Eventually, convincing results from organized trials in Wales [16], Wisconsin [17], and elsewhere were published and confirmed, along with other supportive data on benefits [18]. The Wisconsin study, a randomized clinical trial assessing 650,341 infants during nine years of enrollment demonstrated short- and long-term nutritional benefits with early, aggressive care management [19,20] and less lung disease in those responding well to better nutrition [21]. 

In retrospect, it is much easier to evaluate nutritional outcomes after NBS than the course of lung disease, particularly in children, because of the numerous variables influencing the respiratory system as illustrated in Figure 4, including environmental factors such as respiratory pathogen exposures that are difficult to quantitate. The lesson learned from this experience is that for CF NBS to be *worthwhile*, expeditious follow-up care must ensure that not only will high-quality sweat testing be provided promptly to confirm diagnoses but that the nutritional benefits are achieved immediately by a team of dedicated, experienced caregivers with gastrointestinal/nutritional expertise. Thus, regional data tracking methods should assess multiple indices of nutritional status and monitor growth velocity carefully during the first two years of life. If growth failure occurs, comprehensive assessments and more aggressive nutritional interventions are essential, but more research is needed on supplements such as essential fatty acids. 

Management of the respiratory complications of CF after NBS is more challenging with conventional therapies. Early diagnosis provides the opportunity to initiate some prophylactic measures, monitor for signs and symptoms of lung disease, and intervene quickly. A lesson learned in the Wisconsin trial is that infants must be segregated from older patients to avoid exposure to virulent respiratory pathogens [22]. Although cohorting non-infected patients was not the standard practice during the 20th century, this preventive strategy is essential to make CF NBS *worthwhile* [23,24]. Thus, regions contemplating the initiation of a screening program need to organize a segregated follow-up system for patients diagnosed as neonates. In addition, if/when more CFTR modulator therapy options become approved for infants with the p.Phe508del variant and routine organ preservation a routine reality, all regions will need to ensure the availability of these expensive therapies. In addition, access to care and avoidance of disparities needs to be assured. 

With regard to the risks of CF NBS, many investigations worldwide have focused on potential psychosocial harms [25,26], especially in false-positive families, but it should be recognized that these risks accompany every screening test whether infants or older individuals are being screened. However, some have argued that CF NBS deserves more attention than that given to other conditions, and certainly there have been more studies on the potential risks. Recognizing the importance of the benefit: risk relationship, the Wisconsin team conducted their trial as a comprehensive longitudinal project that assessed adverse outcome potential through a variety of psychosocial studies and interviews [27,28]. The advantages and challenges of genetic counseling were also investigated and generally shown to be beneficial [29]. In summary, the lesson learned is that psychosocial harms may indeed occur, primarily due to misunderstanding of test results and their implications, but that investment by the follow-up team in proactive, excellent communication efforts can prevent and/or alleviate this risk which applies particularly to families experiencing false-positive tests. The importance of such tactics is underscored when a baby is identified as having CRMS/CFSPID (cystic fibrosis transmembrane conductance regulator-related metabolic syndrome/cystic fibrosis screen positive, inconclusive diagnosis) [30]. This condition is considered a “byproduct” of CF NBS with the IRT tier and often presents a diagnostic dilemma. In these cases, and whenever, CF is diagnosed, genetic counseling is essential and should be an integral part of the follow-up efforts [29]. 

## 3. Criteria to Implement Screening 

### 3.1. European CF Society Guidelines

The European Cystic Fibrosis Society (ECFS) has published best practice guidelines for CF NBS [31] that are applicable to the question *Where is it worthwhile*? These deal with population characteristics such as the incidence of CF in a given region, the health and social support resources that are “minimally acceptable for newborn screening to be a valid undertaking,” the quality of dried blood samples, acceptable levels of sensitivity and specificity, and the importance of timeliness. Table 2 summarizes the ECFS recommendations. 

We respectfully disagree with the view that an incidence of at least 1:7000 hould guide decisions about whether or not to screen. In fact, many genetic disorders included in standard NBS panels have a much lower incidence without any challenge to their validity [32]. Many of the hereditary metabolic disorders in NBS panels have incidences of 1:200,000–3,000,000, as David et al. [32] emphasize. In addition, with CF NBS underway for extensive periods, the incidence of CF may decrease significantly [33] and even in previously high incidence regions become less than 1:7000. Certainly, these regions should not discontinue their programs. In view of the wide range of CF incidence data in countries with sufficient CF prevalence to warrant CF care centers and the reduction potential of prenatal and neonatal screening, we are reluctant to specify an incidence criterion; however, based on the typical panel of hereditary metabolic diseases in current screening programs of the western world, greater than 1:25,000 would be reasonable. It should be emphasized that the criteria established by Wilson and Jungner in 1968 [5] for the implementation of a screening program in the general population did not include the concept of a minimal incidence.

The ECFS recommendation that “programmes should aim for a minimum sensitivity of 95%” is appropriate but unattainable in regions using IRT/IRT or some other combination of biomarkers. Such regions are often limited by inadequate knowledge of the *CFTR* mutations prevalent in their population, but as that information is gained transformation to a more sensitive screening test can be accomplished. 

The issue of follow-up efficiency was also addressed in the ECFS guidelines. First, it is stated that a sweat test result should be completed and the result reported to the family on the same day—an ideal practice but unfortunately not always routinely done. With regard to timeliness, the guidelines recommend 35–58 days after birth for the “maximum acceptable age of an infant on the day they are first reviewed by a special CF team following a diagnosis of CF after NBS.” This recommendation was influenced by practical considerations in various European countries, but undoubtedly infants with CF are susceptible to potentially fatal salt loss in sweat prior to the 35–58-day interval, especially in hot climates and with breastfeeding. In addition, CF infants certainly can develop biochemically severe nutritional abnormalities within 2–4 weeks of birth and even suffer the onset of lung disease within 1–2 months. In addition, the recent data suggesting that organ preservation can be achieved with early CFTR modulator therapy [34] argues for diagnosis as soon after birth as possible. Consequently, a more efficient plan promulgated by some organizations recommends diagnosis as early as 2 weeks of age and definitely by 4 weeks of age. To be *worthwhile*, therefore, regions should organize their CF NBS programs to be highly efficient and avoid any preventable delays. 

### 3.2. Clinical and Laboratory Standards Institute, the Association of Public Health Laboratories, the Centers for Disease Control and Prevention, and the Cystic Fibrosis Foundation

In the United States, the group of organizations listed above has worked collaboratively in the area of CF NBS to ensure expeditious nationwide implementation and ongoing attention to quality improvement. The CLSI recently published new guidelines [35] to revise the recommendations of 2011 focusing on the six aspects of CF NBS listed below as the responsible Document Development Committee identified the key areas of quality improvement.

(1)Reassessed IRT cutoff value guidelines and discussed the use of a floating rather than fixed cutoff value. The floating cutoff strategy using the 95th or 96th percentile helps overcome the seasonal and kit-related variations in IRT [11]. The recommendations included: “Recent data have shown that the traditional IRT cutoff values in the IRT/IRT algorithm were too high to minimize false-negative screening results and the 95th to 97th percentile (approximately 60 ng/mL) should be used.” As expanded genetic analyses and next-generation sequencing are becoming less expensive, some CF NBS programs are operating with a lower fixed IRT (for example 40 ng/mL), thus allowing more samples for genetic testing to reduce false-negative screening results.(2)Revised recommendations regarding *CFTR* variant panels based on the most current information including new biotechnologies such as next-generation sequencing, pointing out that “Guidelines published in 2001 and revised in 2004 include recommendations for screening with a *CFTR* variant panel of 23 disease-causing variants with a prevalence of at least 0.1% in the CF population. Although this recommended panel provides a high CF detection rate... additional variants may need to be added for improved CF detection in other ethnic groups. Many NBS programs use larger *CFTR* variant panels...”(3)Assessed using PAP for detecting babies at risk for CF but did not make a recommendation.(4)Discussed communications strategies related to the detecting of CF heterozygote babies and providing genetic counseling.(5)Reviewed emerging issues related to using genetic and genomic sequencing in NBS.(6)Described the existing CF NBS algorithms, while commenting on the advantages and disadvantages of each protocol.

The U.S. CF Foundation organized a recent diagnosis consensus with international input to (1) clarify the criteria that need to be met for diagnosis via either NBS or after signs/symptoms; (2) emphasize the importance of efficient follow-up of positive screening tests; (3) describe how to apply and communicate genetic data; (4) harmonize the definition of CRMS and CFSPID [36]. These guidelines recommend that sweat testing be performed as soon as possible after 10 days of age, ideally by 28 days of age. They also point out that treatment should not be delayed when sweat testing is unsuccessful. The Association of Public Health Laboratories, through its NewSTEPS program, has also emphasized timeliness. Lastly, the Centers for Disease Control and Prevention has established an invaluable quality assurance monitoring program for worldwide assistance *gratis* and a molecular assessment program, which conducts site visits to U.S. NBS laboratories that carry out molecular testing.

For CF NBS to be *worthwhile*, all the guidelines and recommendations summarized above should be well known to the leaders of screening regions and those that wish to implement programs. In the past, too many regions initiated CF NBS programs without taking advantage of the readily available resources and experience of established programs. 

## 4. Incidence of CF around the World and Screening Protocols Being Employed 

Figure 5 and Figure 6 provide data on the estimated incidence of CF in many regions. Through a complete registration of cases directly at birth, the implementation of CF NBS has allowed a more accurate determination of the incidence of CF and better monitoring of its time trends. Before the implementation of NBS, the incidence estimation was mainly based on epidemiological studies that generally suffered from ascertainment bias due to under-diagnosis and/or under-reporting of cases. However, with NBS data, care must be taken when interpreting incidence data, as variations may occur depending on the patients included in the calculations (e.g., false-negatives, patients with meconium ileus, patients with CFSPID). In order to have consistent data, it is important to ensure that the calculations are based on the same population. The incidence data may also be biased by a short observation period in some studies.

### 4.1. Europe

The incidence of CF has long been estimated at 1:2500 in the European population [37]. In 2007, a review of CF NBS programs revealed that the incidence was on average 1:3500 [38] and it appears still lower nowadays. Beyond Ireland which has the highest incidence of CF in Europe (1:1353) [39], the incidence ranges from 1:2800 in the UK [38] to 1:10,000 in Russia [40] (Figure 5). It is 1:2850 in Belgium [41], about 1:4500 in France [42], Germany [14], Italy [38,43,44], and Spain (where large regional variations are observed) [45], while it oscillates between 1:5200 and 1:6500 in Central Europe (Czech Republic [32], Denmark [46], Netherlands [47], Poland [48], Slovakia [49], and Sweden [50]). The incidence appears lower than 1:7000 in three European countries (Portugal [51], Norway [52], and Russia [40]) as well as in various regions of Spain [45]. In countries without NBS, the incidence ranges from 1:2000 (Romania [53]) to 1:25,000 (Finland [54]).

The implementation of CF NBS across Europe has gradually spread, with a faster pace during the past decade. From the update performed by Barben et al. in 2016 [55] and the data acquired since, CF NBS has been implemented in 22 European countries to date (Figure 7). Nineteen countries have a national program and three have regional programs (which cover the whole country for Spain). The number of countries with a national NBS program has gradually increased over the past years, from 2 in 2007 (France and Austria) to 19 to date. Twenty-four countries have no NBS program but nine are considering or planning to implement screening protocols. The screening protocols, however, are varied and all national programs have a distinct algorithm. As illustrated in Figure 7, most programs use DNA analysis as a second-tier test, while five (Austria, Portugal, Russia, Slovakia, and Turkey) still rely exclusively on biochemical tests. The expansion of CF NBS across Europe has been successful and reveals that this screening is considered *worthwhile*.

### 4.2. Australasia

The incidence of CF is well defined in Australasia as CF NBS has been in place for a very long time in this part of Oceania. The incidence is approximately 1:3000. It has been estimated as 1:2821 in New South Wales [56], 1:3139 in Victoria [57], and 1:3180 in New Zealand [58]. Newborn screening for CF has been performed for almost 40 years (1981) in New Zealand—which was the first country to implement a national program (1981) and for almost 20 years in all states within Australia where CF NBS was first introduced in New South Wales during 1981 [56]. Currently, all states use a DNA-based NBS program. The very long experience of Australasia in CF NBS confirms the view that is this screening is deemed *worthwhile* in that part of the world.

### 4.3. United States of America 

The incidence of CF for the entire population is approximately 1:4000, but ethnicity-related variations occur and have a regional impact [59]. In the white population, about 1 in 3000 babies are at risk for CF. Although limited data have been collected and analyzed on “minority” populations, it appears that the incidence and Hispanic infants are about 1:6000 and in African-Americans at least 1:10,000. Disparities in the age of diagnosis and ascertainment of CF cases may occur in these “minority” populations, so more data are needed. It has become increasingly clear that *CFTR* panels need to be expanded to reduce disparities, but NBS labs in the U.S. are notoriously slow to change methodologies as evidenced by a 10-year delay in all states transforming from IRT/IRT to IRT/DNA or IRT/IRT/DNA protocols after unequivocal evidence was published [11]. However, during the current year, all states are using DNA-based CF NBS and all consider their programs *worthwhile.*

### 4.4. Canada

The incidence of CF has been well defined during the past decade in Canada as all 10 provinces implemented CF NBS from 2007, beginning with Alberta, to 2017 when Québec implemented their excellent program based on clinical outcomes. In general, the Canadian population is more Euro-American than in other American countries. Therefore, it is not surprising that the incidence of CF is higher—averaging about 1:3300 [60]. Moreover, Québec in its first two years of CF NBS identified a 1:2300 incidence. The relatively high prevalence of CF throughout Canada and the excellent CF care centers providing follow-up and early treatment have certainly made CF NBS *worthwhile* there using DNA-based protocols.

### 4.5. Latin America—Mexico (North America), Central, and South America

The Latin American population is one of the most diverse in the world, due to a variety of ancestries, ethnicities, and races that have mixed for centuries. The dominant racial groups in the region are Caucasians, European-Amerindian (mestizo), Black, and Amerindian. The proportion of each group varies significantly among the Latin American countries [61]. Given this complex composition, the incidence of CF is difficult to foretell and is further complicated because many countries lack established clinical programs, newborn screening, and registries. 

The racial distribution of CF cases, as illustrated by the 2017 Brazilian Registry, highlights the diversity of CF in Latin America. Of 5128 cases, 68% are white (branca), 25% are mestizo (parda), and 6% are black (preta). There are just a small number of cases of Asian (amarela) and Amerindian (indigena) descent. 

Of 29,887 patients reported by the U.S. Cystic Fibrosis Foundation Patient Registry 2017 Annual Data Report, 8.7% are Hispanics. Extrapolating the current prevalence of CF in the US Hispanic population to Latin America, one could expect about 30,000 people with CF in the region. It should be pointed out that most of the Hispanics in the U.S. come from Mexico and Central America, which has a lower proportion of people of European descent that countries like Argentina, Uruguay, Chile, and Brazil [62]. Thus, the above prevalence might underestimate the number of people with CF.

Currently, reliable data on the incidence of CF in Latin America is lacking, in part, due to limited diagnostic accuracy. There is no neonatal screening in most of the countries, and genetic panels have a low diagnostic yield because they do not reflect the ethnic admixture of the population [63].

#### 4.5.1. Argentina

Although Argentina approved legislation for mandatory administration of neonatal screening for CF in 1994, it has never reached meaningful national coverage (15%–20%). A recent report from the Grupo Registro Nacional de Fibrosis Quistica (National CF Registry Taskforce), verified that the neonatal screening in 2012 had coverage of 28.8%. The CF National Consensus of 2008 reported that between 1995 and 2005, CF incidence was 1:6131, after screening almost 1 million infants. There is no defined protocol for its performance. The most commonly used is IRT/IRT. The Autonomous City of Buenos Aires (CABA) started its neonatal screening in 2002, using the IRT/IRT algorithm. From 2002–2014, the incidence was 1:7444 (49/364,782; V. Rodriguez, personal communication). In 2015, Buenos Aires modified the algorithm to IRT/PAP. 

#### 4.5.2. Brazil

Brazil has developed the most robust CF care structure, similar to the OECD countries, including a CF center network, sophisticated registry, and neonatal screening. Adoption of the neonatal screening (IRT/IRT) has gained prominence in diagnosing new cases, from 32% in 2009 to 61% in 2017. The reported incidence varies among different states, higher in the South Region (Santa Catarina: 1:6500; Paraná: 1:9000), lower in others (Minas Gerais: 1:11,000), (V. Rodriguez, personal communication). 

#### 4.5.3. Chile

Studies from the mid-1990s reported an incidence of 1:4000 (Rios, 1994). More recent data from two pilot studies on neonatal screening puts the incidence between 1:8000 and 1:10,000 (2015–2017; ML Boza, personal communication). Chile anticipates starting national CF neonatal screening in 2020, using the IRT/PAP protocol. 

#### 4.5.4. Mexico

There is no accurate data on incidence. In 2016, the Secretaria de Salud (Ministry of Health) reported 350 new cases of CF per year. Averaging 2.3 million births per year over the last decade, Mexico could have an incidence of approximately 1:6700. Others have reported a lower rate (1:8500; 2002, Jose Luis Lezana, personal communication). Neonatal screening for CF was added to the national program in 2015. The screening method currently used is IRT/IRT. As it has been the norm in the Latin American countries, the use of mutation panels has low yields (A 34-mutation panel had a sensitivity of <75%; Orozco, 2000).

#### 4.5.5. Uruguay

Uruguay implemented neonatal screening in 2010. Initially using the IRT/IRT protocol, but changing to IRT/PAP in 2012. Analysis from years 2010– 2016 totaling 322,727 screened babies, 39 confirmed diagnoses, puts the incidence in 1:8300 (C Pinchack, personal communication).

#### 4.5.6. Other Latin American Countries

In 2019 Colombia and Peru approved the addition of CF screening into their national newborn screenings. We have no information on the protocols used. Also, the incidence in these countries has not been established. Costa Rica has had CF neonatal screening for several years but has not reached full coverage. Seven cases on average per year suggest an incidence of 1:15,000 (J. Gutierrez, personal communication). An ongoing pilot study puts the rate in 1:10,000. 

#### 4.5.7. Is CF Neonatal Screening *Worthwhile* in Latin American Countries?

With an expected high number of cases and the late age in diagnosis that presently occurs, the use of neonatal screening should be a cost-effective tool to help to identify patients at a younger age, therefore improving survival.

### 4.6. Asia 

Although the incidence of CF in Asia has long been unknown, the existence of CF in that continent is now well established. The incidence is greatly variable and appears much higher in the Middle East than in East Asia (Figure 6). Cases of CF have been reported in many Arab countries and in some of them (where consanguinity is common), the incidence appears close to that observed in populations of European descent. It has thus been estimated to 1:2560 in Jordan [64] or to 1:5800 in Bahrain [65], while it is about 1:16,000 in the United Arab Emirates [66] and in Israel, where the incidence has dropped significantly following implementation of population carrier screening program [67]. The incidence of CF is lower in South Asian populations. It has been estimated between 1:10,000 and 1:100,000 in the Indian population [68,69,70,71] and to 1:90,000 in an Oriental population living in Hawaii [72]. The incidence is very low in Japan (1:350,000) [73] as well as in China where less than 30 cases have been reported over the two last decades [74]. Thus, CF NBS would not seem *worthwhile* in those countries. 

Beyond variability in carrier frequency, the wide range observed in incidence may be in part explained by under-diagnosis and under-reporting of cases [75,76]. The incidence of CF in Asia may therefore be under-estimated. Despite the low incidence of CF in that continent, the number of CF patients must be high in some countries due to the large population size (such as India). 

In Arab countries, the major challenges are to improve the diagnosis and the detection of mutations before irreversible organ damage has developed and to promote the constitution of national registries [75,76], which will help to improve CF management and health policy planning. To the best of our knowledge, no CF NBS program is implemented to date in the Arab world, but CF NBS is promoted by the annual meeting for NBS in the Middle East and North Africa. The sole country that was considering implementing an NBS program during 2020 is Israel. 

In view of the high incidence of CF in some Arab countries and the genomic revolution that is underway in those countries (through the Saudi, Qatar, and Emirati genome projects), CF NBS should be *worthwhile* in the regions where CF is well managed. NBS programs would have to take into account the genetic specificities of that population (limited mutation spectrum, lower prevalence of p.Phe508del mutation, mutations found only in that population [76]) but also the high consanguinity rate. 

## 5. Summary

Newborn screening for CF has been performed in many countries for as long as four decades and has transformed the routine method for diagnosing this genetic disease and improved the quality and quantity of life for people with this potentially fatal disorder. Each region has typically undertaken CF NBS after analysis of the advantages, costs, and challenges, particularly regarding the relationship of benefits to risks. The very fact that all regions that began screening for CF have continued their programs implies that public health and clinical leaders consider early diagnosis through screening to be *worthwhile*. In this article, after summarizing the considerations that led the majority of North American and European countries to implement CF NBS programs successfully, we analyze countries that are or should be planning to screen newborns for this relatively common genetic disorder. From this analysis, we suggest the criteria listed in Table 3. Recent, dramatic advances in therapies offer great promise for all patients diagnosed early and especially children diagnosed before pathology quickly develops irreversibly.

## Figures and Tables

**Figure 1 IJNS-06-00018-f001:**
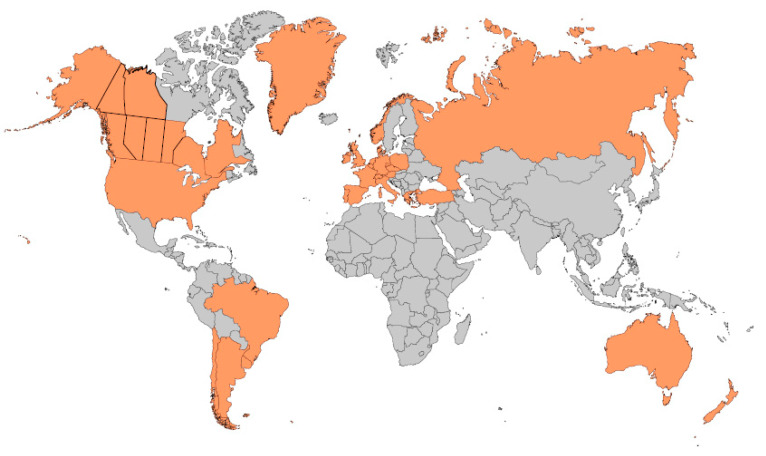
Worldwide implementation of cystic fibrosis newborn screening as of 2020.

**Figure 2 IJNS-06-00018-f002:**
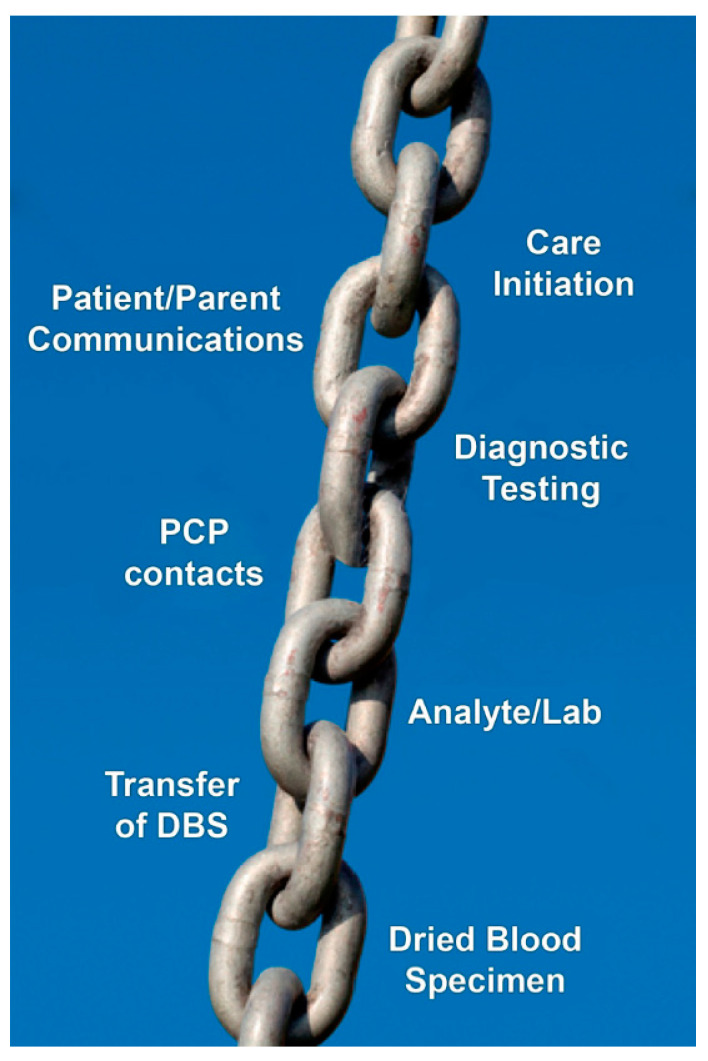
The sequence of processes and procedures linked together like a chain in the system of early diagnosis via newborn screening, reminding us that “a chain is only as strong as its weakest link.” Abbreviations include PCP—primary care provider; DBS—dried blood specimen.

**Figure 3 IJNS-06-00018-f003:**
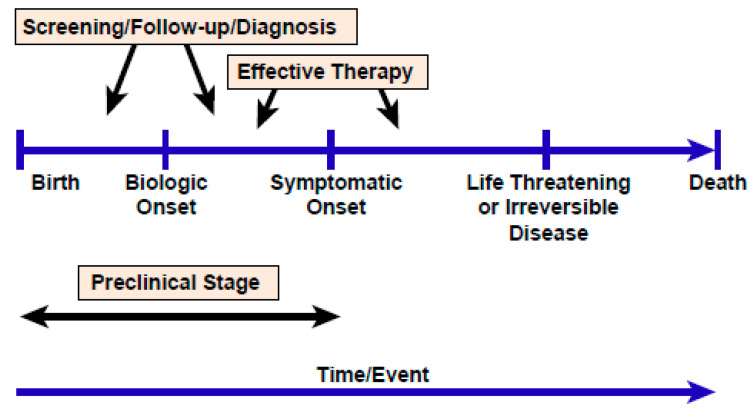
The rationale for early diagnosis via newborn screening by applying the principle inherent in the preventive medicine strategy to detect disease before its symptomatic onset.

**Figure 4 IJNS-06-00018-f004:**
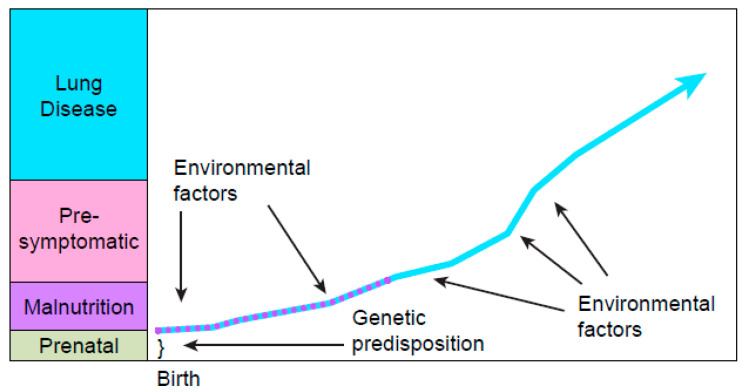
The many intrinsic and extrinsic variables (risk factors) that influence the course of cystic fibrosis and have much more impact on lung disease over a longer time period than those that affect nutritional status. The numerous environmental factors include exposures to smoke, virulent respiratory bacterial pathogens such as mucoid *Pseudomonas aeruginosa*, respiratory virus epidemics, etc.

**Figure 5 IJNS-06-00018-f005:**
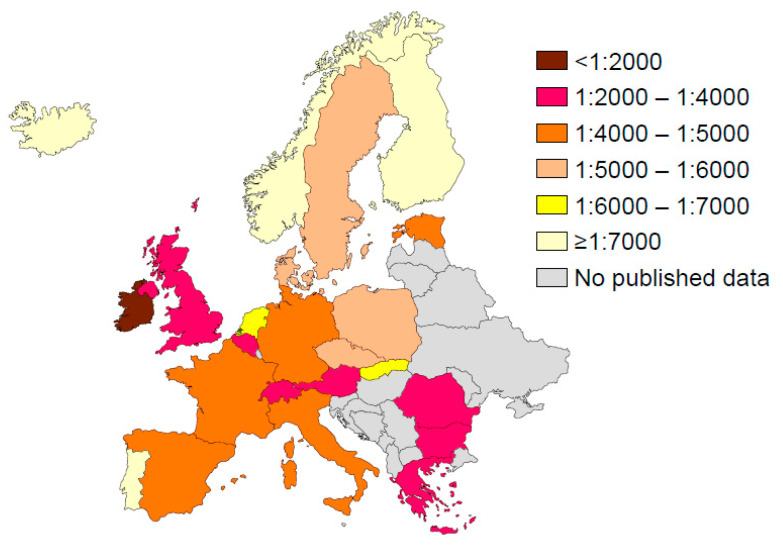
Incidence of cystic fibrosis in Europe.

**Figure 6 IJNS-06-00018-f006:**
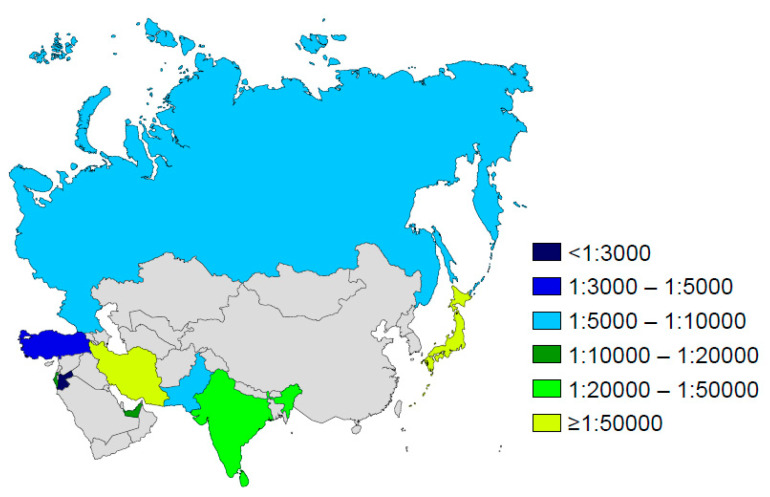
Incidence of cystic fibrosis in Asia.

**Figure 7 IJNS-06-00018-f007:**
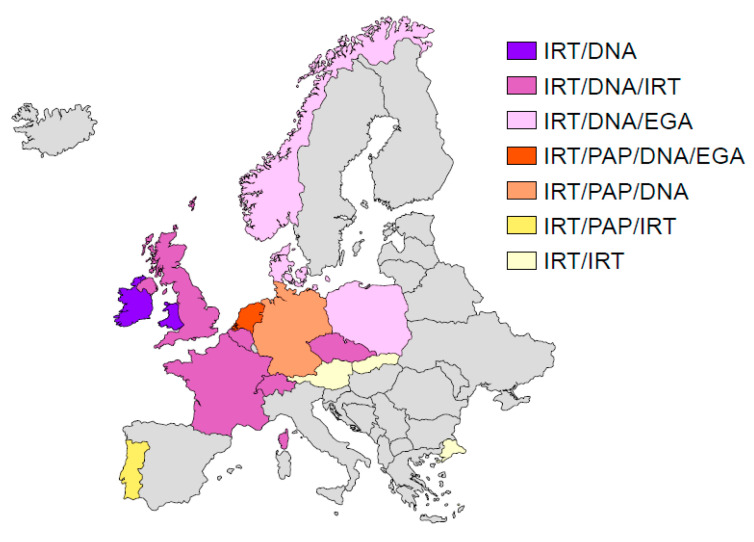
Protocols used in newborn screening for cystic fibrosis in Europe in 2020. Abbreviations include IRT—immunoreactive trypsinogen; EGA—expanded or extended gene analysis; PAP—pancreatitis-associated protein.

**Table 1 IJNS-06-00018-t001:** Essential elements to ensure that cystic fibrosis newborn screening is *worthwhile*.

1	A system must be established and functioning well for the universal collection of dried blood spot specimens and their analysis in a central laboratory with quality assurance mechanisms in place and a goal to maximum sensitivity with acceptable specificity.
2	Collaborative efforts by a team that includes NBS laboratory leadership and CF center follow-up clinicians organized to operate efficiently.
3	Effective CF NBS analytical tests organized as a sequential protocol (algorithm) to maximize sensitivity and optimize specificity.
4	Quality improvements in laboratory methods must be planned for and implemented as technologies advance rather than accepting the *status quo* and resisting change.
5	Expeditious follow-up care must ensure that not only will high-quality sweat testing be provided promptly to confirm diagnoses but that the nutritional benefits are achieved immediately by a team of dedicated, experienced caregivers with gastrointestinal/nutritional expertise.
6	A cohort follow-up system must be ensured for patients diagnosed as neonates to segregate them from older patients and avoid exposure to virulent respiratory pathogens.
7	To ensure a favorable benefit: risk relationship, preventive management of potential psychosocial harms must be given priority by a skilled, dedicated follow-up team.
8	The incidence of CF must be high enough to warrant CF care centers in the NBS region.
9	The NBS system must be organized as a highly efficient operation that avoids preventable delays and ensures consistently diagnostic timeliness.
10	CF NBS guidelines should be known and adhered to throughout the sequence of integrated processes.

**Table 2 IJNS-06-00018-t002:** European Cystic Fibrosis Society (ECFS) best practice guidelines: the 2018 revision [31].

1	Population characteristics that validate screening newborn infants for CF.“Health authorities need to balance the benefit/risk ratio of screening newborns for CF in their population. If the incidence of CF is <1/7000 births, careful evaluation is required as to whether NBS is valid. The protocol must be shown to cause the minimum negative impact possible on the population. Other factors in making the decision on whether to implement screening should include available healthcare resources and the ability to provide a clear pathway to treatment.”
2	Health and social resources that are minimally acceptable for NBS to be a valid undertaking.“Infants identified with CF through a NBS program should have prompt access to specialist CF care that achieves ECFS standards. A NBS program may be a mechanism to better organize CF services, through the direct referral of infants for specialist CF care. Countries with limited resources should consider a pilot study to assess the validity of NBS and the adequacy of referral services for newly diagnosed infants in their population.”
3	Acceptable number of repeat tests required for inadequate dried blood samples for every 1000 infants screened.“The number of requests for repeat dried blood samples should be monitored and should be 0.5%. More than 20 repeats for every 1000 infants, is unacceptable (2%).”
4	Acceptable number of false-positive NBS results (infants referred for clinical assessment and sweat testing).“Programmes should aim for a minimum positive predictive value of 0.3 (PPV is the number of infants with a true positive NBS test divided by the total number of positive NBS tests).”
5	Acceptable number of false-negative NBS results. These are infants with a negative NBS test that are subsequently diagnosed with CF (a delayed diagnosis).“Programmes should aim for a minimum sensitivity of 95%.”
6	Maximum acceptable delay between a sweat test being undertaken and the result given to the family. “The sweat test should be analyzed immediately and the result reported to the family on the same day.”
7	Maximum acceptable age of an infant on the day they are first reviewed by a specialist CF team following a diagnosis of CF after NBS.“The majority of infants with a confirmed diagnosis after NBS should be seen by a specialist CF team by 35 days and no later than 58 days after birth.”
8	Minimum acceptable information for families of an infant recognized to be a carrier of a CF-causing mutation after NBS.Families should receive a verbal report of the result. They should also receive written information to refer to. Information should also be sent to the family Primary Care Physician. The information should be clear that the infant does not have CF; the baby is a healthy carrier; future pregnancies for this couple are not free of risk of CF and the parents may opt for genetic counseling, and there are implications that could affect reproductive decision making for extended family members and the infant when they are of childbearing age.

**Table 3 IJNS-06-00018-t003:** Suggested criteria for cystic fibrosis newborn screening.

Incidence of CF: greater than 1:25,000
Aim at minimum sensitivity of 95%
IRT/DNA—unless unavailable or not feasible
Diagnosis including sweat chloride within 4 weeks of age
Assessment program for tests, including plans for monitoring and updating
Availability of a complete specialist CF team

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
