# Peer review of "Newborn Screening for CF across the Globe—Where Is It Worthwhile?"

_2409-515X, 2020, doi:10.3390/ijns6010018_

Round 1
Reviewer 1 Report
This is a well written manuscript describing newborn screening for CF throughout the world. There is a useful summary of the European CF Society best practice guidelines for CF newborn screening and also guidelines from various groups collaborating in the US. In addition, there some information regarding reported incidence of CF in various countries.
Major comments:
The view of newborn screening for CF is too simplified with very little discussion about the challenges of screening. A discussion of where CF screening is “worthwhile” minimizes the importance of CF screening. With sufficient resources, are there any areas of the world where CF screening is not worthwhile? With data from over 40 years of CF screening, it seems that there are well-established reasons for performing CF screening, so this slant of the manuscript is puzzling. There is little explanation of babies identified as having CRMS/CFSPID. It would be useful to have a short discussion about the challenges associated with this group of infants. Figure 4 – There is little explanation of Fig 4 in the text (e.g. environmental factors)
Minor comments:
P7 line 148 “organized” needs to be corrected
The colors in figures 5 and 6 are difficult to distinguish especially <1:2000 and 1:2000 to 1:4000 in fig 5.
P10 lines 183 and 184: “In fact” used twice.
Author Response
This is a well written manuscript describing newborn screening for CF throughout the world. There is a useful summary of the European CF Society best practice guidelines for CF newborn screening and also guidelines from various groups collaborating in the US. In addition, there some information regarding reported incidence of CF in various countries.
Thank you.
Major comments:
- The view of newborn screening for CF is too simplified with very little discussion about the challenges of screening.
Comments were added in the revised manuscript to describe more of the challenges in the text per se and thereby supplement Table 1.
- A discussion of where CF screening is “worthwhile” minimizes the importance of CF screening. With sufficient resources, are there any areas of the world where CF screening is not worthwhile? With data from over 40 years of CF screening, it seems that there are well-established reasons for performing CF screening, so this slant of the manuscript is puzzling.
We would have preferred the word “essential” to “worthwhile,” but our assignment from the Editors (Drs. Barben and Southern) was to focus on the “worthwhile” concept. The low incidence of CF in some Asian countries and probably Africa, combined with their more basic public health challenges, suggests that these are areas of the world where CF NBS would not be worthwhile.
-There is little explanation of babies identified as having CRMS/CFSPID. It would be useful to have a short discussion about the challenges associated with this group of infants.
We agree but another paper in the IJNS Special Issue by Munck should cover CRMS/CFSPID; however, we added another sentence.
- Figure 4 – There is little explanation of Fig 4 in the text (e.g. environmental factors)
Thank you for pointing out this omission, which is now addressed in the both the figure legend and the text.
Minor comments:
- P7 line 148 “organized” needs to be corrected.
We changed “organized” by “conducted”.
- The colors in figures 5 and 6 are difficult to distinguish especially <1:2000 and 1:2000 to 1:4000 in fig 5.
We improved the colors.
- P10 lines 183 and 184: “In fact” used twice.
We deleted the second “In fact” as not to be redundant.
Reviewer 2 Report
Newborn Screening for CF Across the Globe – Where is it Worthwhile?
Scotet V, Gutierrez H and Farrell M.
The article is a very interesting overview over CF-NBS historically and as a global perspective of CF.
It describes implementation of CF-NBS from the beginning and further on as it is has spread all over the world including difficulties both involving capacity and economy.
The question as to whether it is worthwhile to have CF-NBS has been raised and the authors have tried to address this – with respect to the different perspectives worldwide. It has succeeded with the various different options and availabilities in and within different parts of the world in the majority of countries with a higher incidence of CF.
Parts of the manuscript contains more explanation– especially when it comes to Latin America and South America.
Most of the figures are informative – figure 2 and 3 could be excluded as both are well addressed in the text.
Some revisions are suggested:
In general: programme or program?
Figure 1: Greenland and the Faroe Islands are included in the Danish Kingdom CF-NBS since 2016 - ref 44 and M. Skov: Personal communication
Figure 2: text under fig and as legend – PCP in figure must be spelled out
Table 1: A cohort follow up system must be ensured for patients diagnosed as neonates must to segregated them from older patients and avoid exposure to virulent respiratory pathogens. Red indicate to delete
Figure 3: delete text under figure – same as legend
Figure 4: same as above. Legend: ….that influence the course of CF and have much more impact on lung disease over a longer time. Red indicate to delete
Line 140-141: if/when CFTR modulator therapy becomes approved for infants. Kalydeco available for >6 months of age, and Orkambi from 2 years of age
Line 145: refer to one of the other topics in this specific issue jr.: Psychological impact of NBS for CF
Jane Chudleigh (London, UK)
Figure 6: Suggestion - Use other colours for different incidences. Orange is 1:5000 -1:10,000 in this fig but 1:4000 to
1:5000 in fig 5
Line 262: 1:3000 in Belgium - 1:5600 in Sweden
Line 264: 1:4800 in Denmark ref 44 - 1:2000 in the Faroe Islands
Line 264: Or higher? 1:3600 Bobadilla et al
Line 275: is five countries correct? – or 4?
Figure 7: EGS: to be spelled out – doesn´t seem to occur anywhere else in the text. Or extended gene analysis (EGA)
Line 301: excellent program: based on/due to?
Line 401-2: reorganize sentence – suggestion: before irreversible organ damage has developed
Summary:
Is it possible according to your comments to suggest some criteria for CF-NBS – in a Table? – such as
Incidence of CF: greater than 1: 25,000
Aim at minimum sensitivity of 95%
IRT/DNA – unless unavailable
Diagnosis incl sweat chloride within 4 weeks of age
Assessment program for tests, incl updating
Availability of specialist CF team
Suggestion:
Table with incidences of CF according to the countries in various continents – or according to frequency
Author Response
The article is a very interesting overview over CF-NBS historically and as a global perspective of CF.
It describes implementation of CF-NBS from the beginning and further on as it is has spread all over the world including difficulties both involving capacity and economy.
The question as to whether it is worthwhile to have CF-NBS has been raised and the authors have tried to address this – with respect to the different perspectives worldwide. It has succeeded with the various different options and availabilities in and within different parts of the world in the majority of countries with a higher incidence of CF.
- Parts of the manuscript contains more explanation– especially when it comes to Latin America and South America.
Thank you. Because there is a paucity of literature on Latin America, we included more explanation on those countries.
- Most of the figures are informative – figure 2 and 3 could be excluded as both are well addressed in the text.
Although figures 2 and 3 are well addressed in the text, they seem to us original and explicit. Therefore we think it is important to keep them, especially since the other two reviewers did not object to them and they are so illustrative.
Some revisions are suggested:
- In general: programme or program?
We used the British spelling “programme” all along the manuscript.
- Figure 1: Greenland and the Faroe Islands are included in the Danish Kingdom CF-NBS since 2016 - ref 44 and M. Skov: Personal communication
Thank you. We were not aware of that. Therefore, we added Greenland and the Faroe Islands to the list of European areas that screen at birth (Figure 1).
- Figure 2: text under fig and as legend – PCP in figure must be spelled out
We deleted the text under figure 2 (which corresponded to the title). We also spelled out PCP (primary care provider) and DBS (dried blood specimen) in the figure legend but prefer to keep the illustration concise by retaining these fairly standard abbreviations; readers who do not recognize them will certainly read the figure legend.
- Table 1: A cohort follow up system must be ensured for patients diagnosed as neonates must to segregated them from older patients and avoid exposure to virulent respiratory pathogens. Red indicate to delete.
We revised this statement to make our point clear.
- Figure 3: delete text under figure – same as legend
We deleted the text under figure 3.
- Figure 4: same as above. Legend: ….that influence the course of CF and have much more impact on lung disease over a longer time. Red indicate to delete
We deleted the text under figure 4 and revised the figure legend.
- Line 140-141: if/when CFTR modulator therapy becomes approved for infants. Kalydeco available for >6 months of age, and Orkambi from 2 years of age
We revised to “more CFTR modulator therapy options for infants with the p.Phe508del variant...”
- Line 145: refer to one of the other topics in this specific issue jr.: Psychological impact of NBS for CF. Jane Chudleigh (London, UK)
As suggested by the reviewer, we added the following reference: Chudleigh, J. Psychological impact of newborn screening for CF. Int. J. Neonatal Screen. 2020, In press.
- Figure 6: Suggestion - Use other colours for different incidences. Orange is 1:5000 -1:10,000 in this fig but 1:4000 to 1:5000 in fig 5
As suggested by the reviewer, we improved the colors in the legend of the maps.
- Line 262: 1:3000 in Belgium - 1:5600 in Sweden
We added the incidence of CF in Belgium and the corresponding reference [Lucotte et al. Hum. Biol. 1995]. We also added Sweden in the list of countries where the incidence is close to 1:6,000 and the corresponding reference [Lannefors et al. Resp. Med. 2002].
- Line 264: 1:4800 in Denmark ref 44 - 1:2000 in the Faroe Islands
This paper of Skov et al. (reference 44) estimated the incidence of CF in Denmark at 1:4,866 (26/126 522). Based on the figures in the manuscript (see below) and to be homogeneous with other incidences, the incidence appears to be 1:5,264 (24/126,338) if we consider only the CF patients (n = 22) and the false negatives (n = 2).
“Of the 126 522 newborn infants, 126 338 were screened (99.85%) … Twenty‐six infants (1:4866) were initially reported with two CFTR mutations, of whom 22 were confirmed to have a CF diagnosis. The remaining four were … thus classified as CFSPID. One of the four children was later found to have the two CFTR variants in cis and was reclassified as a carrier of CF. … We encountered two false negatives.”
To take into account these remarks, we changed the sentence “while it is about 1:6,000 in Central Europe (Czech Republic [31], Denmark [44], Netherlands [45], Poland [46], Slovakia [47]), by “while is it oscillates between 1:5,200 and 1:6,500 in Central Europe (Czech Republic [31], Denmark [44], Netherlands [45], Poland [46], Slovakia [47], Sweden [48])”.
- Line 264: Or higher? 1:3600 Bobadilla et al
The 2002 data published by Bobadilla et al. were in some cases preliminary figures.
- Line 275: is five countries correct? – or 4?
Five national NBS programmes rely exclusively on biochemical tests (Austria, Portugal, Russia (the Russian enclave on the Baltic Sea), Slovakia, Turkey).
- Figure 7: EGS: to be spelled out – doesn´t seem to occur anywhere else in the text. Or extended gene analysis (EGA)
We revised this to EGA (expanded or extended gene analysis), as the reviewer suggests, and explained the abbreviation in the figure legend, but prefer the abbreviation for the figure. Readers who do not recognize them will certainly read the figure legend.
- Line 301: excellent program: based on/due to?
Clinical outcome data.
- Line 401-2: reorganize sentence – suggestion: before irreversible organ damage has developed
- Is it possible according to your comments to suggest some criteria for CF-NBS – in a Table? – such as
Incidence of CF: greater than 1: 25,000
Aim at minimum sensitivity of 95%
IRT/DNA – unless unavailable
Diagnosis incl sweat chloride within 4 weeks of age
Assessment program for tests, incl updating
Availability of specialist CF team
Thank you. We like this thoughtful suggestion and have created another table, adding Table 3.
Suggestion:
Table with incidences of CF according to the countries in various continents – or according to frequency
This remark is very interesting. Nevertheless, we think that a map presenting the incidence is more visual and can be more easily compared with the map of countries which have a NBS programme. Many readers will prefer an “at-a-glance” map.
Reviewer 3 Report
The paper gives an excellent historical review of cystic fibrosis (CF) new born screening (NBS) and contributes in a fine way to the ongoing discusson of the benfits and drawbacks of CF-NBS.
I have some minor comments only:
Page 6, lines 130-137. CF has been reported to to be associated with low birth weight in some populations, thus growth velocity could be more useful than for instance classic antrophometry alone (bmi z-scores) when monitoring CF children during the first two years of life. Further the authors could consider to comment on the suppplement of essential fatty acids as a possible intervention early in life for children with CF.
Page 7, 138-167: The figure is fine. However, the authors have not mentioned access or denied access to care. Shared care challenges equitable treatment both in CF- NBS and CF follow up and should be discussed in this section.
Page 10, lines 222-225: I miss a comment on this statement. -Floating vs fixed IRT Levels:
As genetic testing and (new generation) genetic sequencing is becoming less expensive some CF-NBS programmes are operating with a lower fixed IRT ( for example 40 ng/ml), hence allowing more samples for genetic testing to minimize false-negative screening.
Page 12, lines 260-263: The incidence data may also be biased by a short observation period in some of the studies.
Author Response
The paper gives an excellent historical review of cystic fibrosis (CF) new born screening (NBS) and contributes in a fine way to the ongoing discusson of the benfits and drawbacks of CF-NBS.
Thank you.
I have some minor comments only:
- Page 6, lines 130-137. CF has been reported to to be associated with low birth weight in some populations, thus growth velocity could be more useful than for instance classic antrophometry alone (bmi z-scores) when monitoring CF children during the first two years of life. Further the authors could consider to comment on the suppplement of essential fatty acids as a possible intervention early in life for children with CF.
Thank you an interesting observation. Although the LBW association is uncertain, and as you imply apparently true for only “some populations,” we agree that monitoring growth velocity can be valuable and have therefore added a comment. On the other hand, EFA supplements remain to be proven valuable and the dosages determined, so we prefer to describe this as a current research topic, particularly since the Wisconsin team has a major study underway that is revealing much more than we have published previously.
- Page 7, 138-167: The figure is fine. However, the authors have not mentioned access or denied access to care. Shared care challenges equitable treatment both in CF- NBS and CF follow up and should be discussed in this section.
We added a comment on this important topic.
- Page 10, lines 222-225: I miss a comment on this statement. -Floating vs fixed IRT Levels:
We added a comment to explain.
- As genetic testing and (new generation) genetic sequencing is becoming less expensive some CF-NBS programmes are operating with a lower fixed IRT ( for example 40 ng/ml), hence allowing more samples for genetic testing to minimize false-negative screening.
Thank you for this comment which we added to the text.
- Page 12, lines 260-263: The incidence data may also be biased by a short observation period in some of the studies.
The reviewer is totally right. We added this comment to the text.